# Reliability of Spino-Pelvic and Sagittal Balance Parameters Assessed During Walking in Patients with Back Pain

**DOI:** 10.3390/s25061647

**Published:** 2025-03-07

**Authors:** Armand Dominik Škapin, Janez Vodičar, Nina Verdel, Matej Supej, Miha Vodičar

**Affiliations:** 1Department of Orthopaedic Surgery, University Medical Centre Ljubljana, Zaloška cesta 9, 1000 Ljubljana, Slovenia; armand.skapin@kclj.si; 2Faculty of Medicine, University of Ljubljana, Vrazov trg 2, 1000 Ljubljana, Slovenia; 3Faculty of Sport, University of Ljubljana, Gortanova ulica 22, 1000 Ljubljana, Slovenia; janez.vodicar@fsp.uni-lj.si (J.V.); nina.verdel@fsp.uni-lj.si (N.V.); matej.supej@fsp.uni-lj.si (M.S.)

**Keywords:** degenerative spine disease, sagittal balance, gait analysis, dynamic assessment, motion capture

## Abstract

This study aimed to establish and assess the reliability of spino-pelvic and sagittal balance parameters measured during walking in patients with back pain, some of whom had radiological signs of sagittal imbalance, reflecting real-world clinical conditions. Dynamic assessment offers an alternative to conventional static measurements, potentially improving the evaluation of sagittal balance. Ten patients aged 56–73 years completed a six-minute walking assessment while being monitored by the optoelectric Qualisys Motion Capture System. Forty-nine reflective markers were placed to measure the spino-pelvic and sagittal balance parameters across five gait phases: pre-walk, initial-walk, mid-walk, end-walk, and post-walk. Test–retest reliability was evaluated using the intraclass correlation coefficient (ICC). The results showed excellent reliability for thoracic kyphosis angle (ICC = 0.97), C7-L5 sagittal trunk shift (ICC = 0.91), and global tilt angle (ICC = 0.99); good reliability for auditory meatus-hip axis sagittal trunk shift (ICC = 0.85); and moderate reliability for pelvic angle (ICC = 0.57), lumbar lordosis angle (ICC = 0.72), and sagittal trunk angle (ICC = 0.73). Despite minor marker placement inconsistencies and variations in body movement across trials, the findings support the use of this dynamic assessment method in research settings. Its clinical application could also enhance diagnostic accuracy and treatment planning for patients with sagittal balance disorders, allowing for better-tailored therapeutic interventions.

## 1. Introduction

Degenerative spine disease is a prevalent condition among the aging population, often leading to progressive changes in spinal curvatures, particularly the loss of lumbar lordosis due to disk degeneration [1,2]. These changes can significantly impact sagittal balance, which is critical for maintaining an upright posture and ensuring efficient gait biomechanics with minimal muscular effort [3,4,5]. Maintaining sagittal balance is vital for overall mobility and daily functioning, and its impairment is linked to pain and reduced quality of life.

Current clinical evaluation of sagittal balance primarily relies on standing full-spine X-rays, which provide static measurements of spinal alignment. However, these assessments may not accurately reflect a patient’s true sagittal balance, as they fail to capture the dynamic compensatory mechanisms involved in maintaining posture during movement. A key limitation of static imaging is that compensatory mechanisms can temporarily maintain normal sagittal balance during the brief period of X-ray imaging but may fail during dynamic activities like walking [3,4,6,7,8,9,10,11,12]. Therefore, true sagittal balance can sometimes only be observed during physical activity, when muscles of compensatory mechanisms fatigue, leading to a decline in compensated sagittal balance [7,8,9,10,11,12]. Therefore, dynamic assessment of sagittal balance during walking could provide a more comprehensive understanding of spinal function, revealing imbalances that static measurements might overlook.

Several dynamic assessment methods have already been proposed, often utilizing motion capture systems to track anatomical landmarks in real time. However, existing approaches present limitations such as including treadmill-based assessments that do not reflect natural gait patterns, short walking distances, small sample sizes, and the use of single-camera systems, all of which may compromise measurement accuracy and clinical applicability [7,8,9,10,11,12]. Furthermore, most studies focus on isolated parameters rather than providing a holistic assessment of spino-pelvic and sagittal balance parameters [7,8,9,10,11,12].

To address these gaps, this study introduces a novel motion-capture-based method for assessing sagittal balance dynamically during walking in patients with back pain, reflecting real-world clinical conditions. This approach aims to provide a more comprehensive and accurate evaluation of spino-pelvic and sagittal balance parameters by overcoming the limitations of previous methods. Establishing the reliability of this method could enhance its clinical utility by offering a more accurate tool for evaluating sagittal balance, leading to better-informed clinical decision-making and a more personalized surgical approach. By accurately identifying sagittal imbalance, surgeons can tailor interventions to each patient’s needs, ensuring optimal correction of spinal curvatures when surgery is indicated and ultimately improving patient outcomes.

## 2. Materials and Methods

### 2.1. Patients

Participants were screened and enrolled at an orthopedic spine surgery outpatient unit of a single university medical center. The selection process was conducted over a three-month period, during which eligible patients were evaluated based on predefined inclusion and exclusion criteria by the lead researcher. Out of approximately 1200 patients assessed, 85 met the enrollment criteria, and 10 patients ultimately agreed to participate and completed the full study protocol. This study is a part of a larger research project, where establishing the reliability of the measurements is a crucial initial step to ensure consistent further results.

The inclusion criteria were (1) back pain as the main symptom and (2) age between 50 and 80 years. Exclusion criteria included (1) symptoms of spinal stenosis and radiculopathy (neurogenic claudication or buttock and leg pain), (2) previous instrumented spinal surgery, (3) a coronal Cobb angle greater than 30°, (4) symptoms of hip or knee arthrosis, (5) symptoms of vascular intermittent claudication, (6) cardio-pulmonary disease that impairs the patient’s ability to walk for six minutes, and (7) neuromuscular disease.

A total of 10 participants (9 females, 1 male) meeting the eligibility criteria were enrolled in the study. The participants had a mean age of 65.6 ± 5.6 years (range: 56–73 years), an average height of 163.1 ± 8.7 cm, and an average body mass of 73.5 ± 15.4 kg. Participants reported a back pain severity during gait of 4.0 ± 2.1, as measured by the Visual Analog Scale (VAS). Upon enrollment, all participants underwent standing full-spine sagittal X-rays, which were analyzed to measure standard spino-pelvic and sagittal balance parameters. The average sagittal vertical axis (SVA) was 20.4 ± 27.3 mm, and the average pelvic incidence-lumbar lordosis mismatch (PI-LL) was 0.5° ± 9.9°. Notably, four participants had an SVA greater than 40 mm or a PI-LL greater than 10°, suggesting radiological evidence of sagittal imbalance. The demographic and clinical characteristics of the participants are presented in Table 1. Compared to other studies evaluating dynamic spino-pelvic and sagittal balance parameters, our sample is demographically representative [7,8,9,10,11,12]. The mean age aligns closely with previously studied populations, and the predominance of female participants is consistent with published research. The only notable difference is that the BMI in our study is slightly higher than in some prior studies.

### 2.2. Motion Capture System and Marker Placement

Gait analysis was conducted using the optoelectric Motion Capture System (Qualisys AB, Göteborg, Sweden), which was equipped with 12 Oqus 7+ cameras strategically positioned around the testing area to ensure comprehensive data acquisition. Reflective markers were placed directly on the participants’ skin following the standardized “Qualisys PAF package: Instituti Ortopedici Rizzoli (IOR)” protocol. This protocol involves the placement of 49 reflective markers across the participants’ limbs and bodies, enabling a detailed assessment of whole-body kinematics during gait (Figure 1) [13,14,15]. The markers were placed on the following anatomical landmarks, with corresponding labels from Figure 1 in brackets: the glabella (SGL), just above both ears (L_HEAD, R_HEAD), the jugular notch of the sternum (SJN), the xiphisternal joint (SXS), spinous processes of C7 (CV7), Th2 (TV2), L1 (LV1), L3 (LV3), L5 (LV5), the vertebra at the inferior edge of the scapulae (MAI), the acromial edges (L_SAE, R_SAE), the lateral midpoint of the humeri (L_HUM, R_HUM), the lateral epicondyle of the humeri (L_HLE, R_HLE), the styloid processes of the radii (L_RSP, R_RSP) and ulnae (L_USP, R_USP), the base of the forefinger (L_HM2, R_HM2), the anterior and posterior superior iliac spines (L_IAS, R_IAS, L_IPS, R_IPS), the lateral side of the greater trochanter (L_FTC, R_FTC), the lateral and medial femoral epicondyles (L_FLE, R_FLE, L_FME, R_FME), the proximal tip of the fibular head (L_FAX, R_FAX), the most anterior border of the tibial tuberosity (L_TTC, R_TTC), the lateral and medial malleolar prominences (L_FAL, R_FAL, L_TAM, R_TAM), the Achilles tendon insertion on the calcaneus (L_FCC, R_FCC), and the dorsal margins of the first, second, and fifth metatarsal heads (L_FM1, R_FM1, L_FM2, R_FM2, L_FM5, R_FM5).

To maintain consistency, all markers were applied by the same trained individual, who used anatomical palpation techniques to accurately position the markers over bony prominences. During the testing procedure, participants walked barefoot and wore only underwear.

### 2.3. Walking Protocol

To ensure uniform preparation, all participants first completed a standardized five-minute stretching routine led by a kinesiologist. Following this, reflective markers were applied while participants were seated. The walking protocol consisted of continuous walking for six minutes between two cones placed seven meters apart, creating a controlled environment. Participants walked at a normal, self-selected pace. Each participant completed two consecutive measurement sessions, separated by a one-hour rest period. During the rest period, the markers were removed and reapplied before the second measurement.

To capture a detailed dynamic profile, gait data were recorded across five phases as follows:Pre-walk measurement: Before beginning the walking protocol, participants stood still to establish a baseline static posture, simulating the position used during standing full-spine sagittal X-rays. In this position, they held a fabric band in both hands, draped around their necks, replicating the posture used during X-ray imaging, where a calibration ball is held on the band. This positioning resulted in slight shoulder anteflexion and elbow flexion.Initial-walk measurement: The first 30 s of walking were recorded to capture the transition from static to dynamic movement. For analysis, three consecutive steps (each defined as movements between two consecutive heel strikes) were selected from the straight walking segment between the cones, avoiding turns, and ensuring no marker data loss.Mid-walk measurement: Data were collected during a 30 s interval at the midpoint of the walking period. Three consecutive steps without marker data loss, selected from the straight walking segment between the cones, were used for analysis.End-walk measurement: The final 30 s of walking were recorded to assess any changes in gait that might have developed during walking. Three consecutive steps without marker data loss were chosen from the straight walking section between the cones for analysis.Post-walk measurement: After completing the six-minute walking period, participants stood still once again for a final static measurement, holding the fabric band in the same manner as the pre-walk measurement, replicating the posture used during X-ray imaging.

### 2.4. Spino-Pelvic and Sagittal Balance Parameters

The following spino-pelvic and sagittal balance parameters were measured during the five recorded phases:Pelvic angle (PA): The angle between the line connecting the midpoint of the markers placed on the posterior superior iliac spines and the midpoint of the markers on the anterior superior iliac spines and the horizontal plane (Figure 1: markers labeled L_IPS, R_IPS, L_IAS, R_IAS; Figure 2a).Lumbar lordosis angle (LLA): The angle between the lines connecting the markers placed on the spinous processes of lumbar vertebrae 1 (L1) and 3 (L3), and the spinous processes of L3 and lumbar vertebrae 5 (L5) (Figure 1: markers labeled LV1, LV3, LV5; Figure 2b).Thoracic kyphosis angle (TKA): The angle between the lines connecting markers placed on the spinous process of cervical vertebra 7 (C7) and the spinous process of the vertebra at the inferior edge of the scapulae, and the spinous process of the vertebra at the inferior edge of the scapulae and L1 (Figure 1: markers labeled CV7, MAI, LV1; Figure 2c).C7-L5–sagittal trunk shift (C7-L5-STS): The distance between the vertical line passing through the marker on the spinous process of C7 and the marker on the spinous process of L5 (Figure 1: markers labeled CV7 and LV5; Figure 2d).Auditory meatus-hip axis–sagittal trunk shift (AM-HA-STS): The distance between the vertical line passing through the midpoint of the markers placed just above the ears and the midpoint of the markers placed on the greater trochanters (Figure 1: markers labeled R_HEAD, L_HEAD, R_FTC, L_FTC; Figure 2e).Sagittal trunk angle (STA): The angle between the vertical line and the line connecting the midpoint of the markers placed just above the ears and the midpoint of the markers placed on the greater trochanters (Figure 1: markers labeled R_HEAD, L_HEAD, R_FTC, L_FTC; Figure 2f).Global tilt angle (GTA): The angle between the lines connecting markers placed on the spinous process of C7 and L5, and the spinous process of L5 and the midpoint of markers placed on the greater trochanters (Figure 1: markers labeled CV7, LV5, R_FTC, L_FTC; Figure 2g).

All parameters were calculated in a three-dimensional space to account for spinal motion in the sagittal, axial, and coronal planes during gait. For LLA, TKA, C7-L5-STS, AM-HA-STS, STA, and GTA, calculations were performed within a plane defined by markers placed on the xiphisternal joint, jugular notch, and the spinous processes of C7, Th2, L1, L3, L5, and the vertebra at the inferior edge of the scapulae (Figure 1: markers labeled SXS, SJN, CV7, TV2, LV1, LV3, LV5, and MAI). These parameters were determined by projecting the respective markers onto this plane, which moved synchronously with the trunk during walking, capturing its three-dimensional motion. In contrast, PA was calculated relative to the floor, also accounting for body rotations in all three planes.

These measured parameters were selected to closely correspond with established spino-pelvic and sagittal balance parameters typically measured in X-ray imaging. Specifically, PA correlates with sacral slope (SS), LLA with lumbar lordosis (LL), TKA with thoracic kyphosis (TK), C7-L5-STS with sagittal vertical axis (SVA), AM-HA-STS with center of the auditory meatus to hip axis (CAM-HA), STA with odontoid hip axis (OD-HA), and GTA with global tilt (GT) [3,16,17].

### 2.5. Data Analysis

The reliability of the measured spino-pelvic and sagittal balance parameters was evaluated using both inter-trial and intra-trial comparisons. A trial was defined as one completed six-minute walking assessment for a single participant, with each participant performing two trials.

The inter-trial comparison was conducted by comparing measurements obtained during a single step across separate trials within the same walking phase (initial-walk, mid-walk, and end-walk measurements). This analysis aimed to identify potential inconsistencies arising from factors such as marker placement variability and differences in participant movement patterns between trials. Since markers were removed and reapplied between trials, this comparison provided valuable insights into the influence of marker placement inconsistencies.

The intra-trial comparison involved assessing the consistency of measurements taken from two consecutive right or left steps within the same trial and measurement phase. This analysis provided insights into fluctuations in a participant’s body movement over short walking distances, independent of inter-trial variability.

Marker placement consistency was assessed by calculating the distances between specific marker pairs on the same limb in each trial and determining the difference between these measurements across both trials. This difference represents the marker placement error for each pair, reflecting variability due to marker reapplication.

### 2.6. Statistical Analysis

The reliability of the measured parameters was assessed using the intraclass correlation coefficient (ICC 3.1) and ICC (2,k), which are widely used statistical measures for evaluating the consistency of repeated measurements. ICC quantifies the proportion of total variance that is attributable to true differences between measurements, rather than measurement error. ICC values range from 0 to 1, with higher values indicating greater measurement consistency. An ICC of ≥0.90 indicates excellent reliability, values between 0.75 and 0.89 represent good reliability, values between 0.50 and 0.74 indicate moderate reliability, and values < 0.50 indicate poor reliability. In addition, the mean bias and 95% limits of agreement (LoA) were calculated [18]. All statistical analyses were performed using the MATLAB software (version R2020b, MathWorks Inc., Natick, MA, USA).

## 3. Results

The mean values and standard deviations of the measured spino-pelvic and sagittal balance parameters for all participants during two static measurements (pre-walk and post-walk phase) across both trials are presented in Table 2a,b. The data show that most parameters exhibited minimal differences between trials, with some demonstrating greater consistency than others. Among the parameters, TKA was the most stable, with pre-walk values of 29.3° ± 6.0° and 28.9° ± 6.5°, and post-walk values of 29.1° ± 6.4° and 29.0° ± 7.1° for Trials 1 and 2, respectively. In contrast, PA displayed the greatest variability, with pre-walk values of 4.4° ± 2.9° and 4.5° ± 3.3°, and post-walk values of 4.0° ± 2.9° and 3.5° ± 3.4° for Trials 1 and 2, respectively.

The mean values and standard deviations of the measured parameters for all participants during each step of the three dynamic measurements across both trials are presented in Table 2 (c: initial-walk, d: mid-walk, e: end-walk). The results demonstrate slight variations in the measured values between trials for each phase, suggesting overall consistency. Among the parameters, TKA exhibited the highest consistency, with initial-walk values of 30.2 ± 5.8° and 30.5° ± 6.2°, mid-walk values of 29.8° ± 5.9° and 29.8° ± 6.2°, and end-walk values of 29.7° ± 5.6° and 30.3° ± 6.7° for Trials 1 and 2, respectively. In contrast, AM-HA-STS showed the greatest variability, with initial-walk values of 71.2 ± 38.6 mm and 77.5 ± 36.4 mm, mid-walk values of 68.5 ± 29.2 mm and 74.2 ± 30.0 mm, and end-walk values of 65.0 ± 34.3 mm and 78.0 ± 37.5 mm for Trials 1 and 2, respectively.

The test–retest reliability assessed across two trials (inter-trial comparison), as reflected in the intraclass correlation coefficient (ICC), is shown in Table 3. The ICC values for the inter-trial comparison (Table 3 and Figure 3) revealed moderate reliability for PA, LLA, and STA (0.57, 0.72, 0.73, respectively); good reliability for AM-HA-STS (0.85); and excellent reliability for TKA, C7-L5-STS, and GTA (0.97, 0.91, 0.99, respectively). The test–retest reliability within the same trial, assessed between two consecutive right or left steps (intra-trial comparison), as reflected in the ICC, is also reported in Table 3. The ICC values for intra-trial comparison (Table 3 and Figure 3) revealed good reliability for PA, LLA, C7-L5-STS, AM-HA-STS, and STA (0.87, 0.77, 0.89, 0.86, 0.79, respectively) and excellent reliability for TKA and GTA (0.98, 0.99, respectively).

The Bland–Altman plots for both inter-trial and intra-trial comparisons are presented in Figure 4. The plots on the left (intra-trial comparisons) generally exhibit narrower limits of agreement compared to those on the right (inter-trial comparisons), indicating greater consistency when comparing measurements within the same trials. GTA demonstrated the smallest discrepancies in both intra- and inter-trial comparisons, with limits of agreement ranging from −2.83° to 2.41° and −3.07° to 6.03°, respectively. In contrast, LLA exhibited the largest discrepancies in both comparisons, with limits of agreement ranging from −8.94° to 7.19° for intra-trial and from −9.63° to 11.24° for inter-trial comparisons. The greatest increase in variability between intra- and inter-trial comparisons was observed for PA, where the limits of agreement widened from −2.53° to 2.38° in the intra-trial comparison to −5.12° to 5.55° in the inter-trial comparison. On the other hand, C7-L5-STS exhibited the smallest increase in variability, with limits of agreement changing from −21.61 to 21.01 mm in the intra-trial comparison to −28.19 to 23.88 mm in the inter-trial comparison.

Table 4 presents the differences in distances between specific marker pairs placed on the same limb across both trials, along with the calculated average values for each pair. The data indicate generally consistent marker placement, with most differences falling within the range of 0 to 1.9 cm. However, larger discrepancies were observed at certain anatomical sites, particularly between the markers positioned on the greater trochanter and the femoral lateral epicondyle, where differences reached up to 4.4 cm.

## 4. Discussion

Sagittal balance is a complex phenomenon that requires comprehensive diagnostics, with standardized standing full spine X-rays providing only static measurements [1,3,4]. To achieve a more detailed, dynamic evaluation of sagittal balance, we utilized a reflective marker-based motion capture system during walking. While several researchers have proposed protocols for measuring spino-pelvic and sagittal balance parameters dynamically [7,8,9,10,11,12], only Miscusi et al. [19] have assessed their reliability. However, their study was limited to three parameters, included only healthy subjects, and covered a short walking distance of 10 m. Other studies measuring sagittal balance dynamically have also encountered various limitations, such as reliance on treadmill walking, which can alter natural gait patterns, or the use of short walking distances that may not fully capture the effects of fatigue on sagittal balance [7,8,9,10,11,12].

Our method addresses these limitations by measuring a broader set of parameters over a longer walking distance on solid ground, providing a more comprehensive and clinically relevant evaluation of sagittal balance dynamics. The longer walking distance better simulates muscle fatigue and dynamic (intermittent) clinical symptoms, which typically worsen with increased muscle demand. Furthermore, by including participants with back pain and radiological signs of sagittal imbalance, we enhanced the study’s clinical relevance, reflecting real-world conditions where spino-pelvic and sagittal parameters are typically evaluated. The primary objective of this study was to assess the reliability of seven parameters measured over an extended walking period (six minutes) in individuals with back pain.

Reliability concerns for this walking assessment may arise due to variations in participants’ body movements between consecutive steps, inconsistencies across different trials, and potential human error in marker placement. The Bland–Altman plots (Figure 4—left side) for intra-trial comparisons, which compare measurements from two consecutive right or left steps within the same trial, reveal notable inconsistencies in body movements over short walking distances. The largest discrepancies were observed in lumbar lordosis angle (LLA), with limits of agreement ranging from −8.94° to 7.19° and an average value of 9.0° ± 5.3° across all dynamic measurements. In contrast, the smallest discrepancies were observed in global tilt angle (GTA), with limits of agreement ranging from −2.83° to 2.41° and an average value of 48.1° ± 12.9°. These findings indicate considerable variability in certain body movements between consecutive steps; however, the overall reliability of the parameters was generally good, with TKA and GTA demonstrating excellent reliability (Table 3, Figure 3).

Comparisons across trials (inter-trial comparison, Figure 4—right side) showed even more pronounced discrepancies compared to intra-trial comparisons (Figure 4—left side). This increased variability can be attributed to inconsistencies in body movement across trials and potential marker placement errors. The most significant increase in discrepancies was observed in pelvic angle (PA), where the limits of the agreement nearly doubled from −2.53° to 2.38° (intra-trial) to −5.12° to 5.55° (inter-trial), with an average value of 6.1° ± 2.2°. In contrast, C7-L5 sagittal trunk shift (C7-L5-STS) exhibited the smallest increase in discrepancies, with limits widening from −21.61 mm to 21.01 mm (intra-trial) to −28.19 mm to 23.88 mm (inter-trial), with an average value of 77.9 ± 22.6 mm. By subtracting intra-trial discrepancies from inter-trial discrepancies, we could isolate the true differences due to repeated assessments, excluding the natural sway of the body between consecutive steps. Despite these differences, ICC values indicated excellent reliability for TKA, C7-L5-STS, and GTA; good reliability for AM-HA-STS; and moderate reliability for the remaining parameters (Table 3).

To evaluate potential marker placement errors, we analyzed the differences in distances between specific marker pairs placed on the same limb across trials (Table 4). These pairs were selected to ensure no major joints were positioned between them, minimizing variability due to joint movement. The results showed that the greatest inconsistencies occurred between the pair of markers on the greater trochanter and the femoral lateral epicondyle, with average differences of 1.4 cm on the left and 1.5 cm on the right. This variability is likely due to the greater trochanter’s relatively large surface area and the substantial soft tissue covering it. However, when assessing parameters defined by these markers (AM-HA-STS, STA, GTA), the increase in discrepancies between intra- and inter-trial comparisons remained relatively low (Table 3, Figure 3), suggesting that marker placement errors did not significantly impact overall measurement reliability.

The major findings of this study demonstrate that the test–retest reliability of the measured parameters (inter-trial comparison) was excellent for TKA (ICC = 0.97), C7-L5-STS (ICC = 0.91), and GTA (ICC = 0.99); good for AM-HA-STS (ICC = 0.85); and moderate for PA (ICC = 0.57), LLA (ICC = 0.72) and STA (ICC = 0.73) (Table 3). The excellent reliability of TKA, C7-L5-STS, and GTA suggests that these measurements can be consistently reproduced, making them valuable for clinical and research applications. Although PA, LLA, and STA demonstrated moderate reliability, their reproducibility remains sufficient for research purposes. Overall, the observed levels of reliability confirm that the method provides consistent results and can be effectively utilized in future studies.

Compared to Miscusi et al. [19], who also assessed the reliability of dynamic spino-pelvic and sagittal balance parameters, our findings revealed some differences. Their study reported ICC values for pelvic tilt angle (defined as the angle between the horizontal axis and a line from posterior-superior iliac spines to anterior-superior iliac spines), trunk angle (the angle between the vertical axis and a line connecting spinous processes of C7 and S1), and sagittal trunk shift (the distance between vertical lines passing through the spinous processes of C7 and S1) during walking, with reliability ranging from moderate to excellent (ICC = 0.867, ICC = 0.526, ICC = 0.953, respectively). They concluded that while trunk angle lacked reliability (ICC < 0.70), pelvic tilt angle and sagittal trunk shift demonstrated high reliability. In our study, lower ICC values were observed for PA, corresponding to their pelvic tilt angle (0.57), and for C7-L5-STS, aligning with their sagittal trunk shift (0.91), which may be due to differences in participant demographics. Their trunk angle does not correlate with any of our measured parameters. Miscusi et al.’s study involved young, healthy volunteers aged 19–32, whereas our participants were older, experiencing back pain, and included individuals with radiological signs of sagittal imbalance, which may have influenced the reliability outcomes.

To our knowledge, Miscusi et al. [19] published the only study evaluating the reliability of motion capture system measurements for spino-pelvic and sagittal balance parameters during gait. Therefore, caution is advised when interpreting findings from other studies [7,8,9,10,11,12] that did not evaluate measurement reliability, as this may impact the overall trustworthiness of the reported conclusions.

Although our study had a relatively small sample size, a substantial number of comparisons were conducted to assess reliability: 90 for inter-trial reliability, where each step from each dynamic measurement was compared across both trials for all 10 participants, and 60 for intra-trial reliability, where two consecutive right or left steps were compared within each measurement across three phases and two trials. To enhance the study design, achieving a more balanced male-to-female ratio would improve the generalizability of the findings. Additionally, conducting the study with different individuals placing the markers may provide further insights into potential marker placement errors in real-world settings. Assessing inconsistencies in body movements across trials, independent of other influencing factors, could be achieved by having participants repeat the walking trials without removing and reapplying the markers. However, this step was considered unnecessary within the scope of this study’s reliability assessment.

Establishing the reliability of this method provides a strong foundation for its implementation in both research and clinical settings. In research, this approach enables the exploration of dynamic changes in spino-pelvic and sagittal balance parameters during walking, providing valuable insights beyond static X-ray assessments. Additionally, future studies could compare these dynamic measurements with radiographic findings and patient-reported outcomes to identify predictive factors for the deterioration of sagittal balance during movement. Clinically, the ability to accurately assess a patient’s true sagittal balance status could significantly enhance surgical planning. By determining if sagittal alignment is maintained or deteriorates during walking, surgeons can tailor procedures to each patient, ensuring optimal correction of spinal curvatures when surgery is indicated. This personalized approach may lead to better postoperative outcomes and improved patient satisfaction.

## 5. Conclusions

The newly introduced method demonstrated excellent to moderate reliability for the dynamic measurement of spino-pelvic and sagittal balance parameters in older participants with back pain. Notably, excellent reliability was observed for thoracic kyphosis angle (TKA), C7-L5—sagittal trunk shift (C7-L5-STS), and global tilt angle (GTA). The high consistency of these measurements underscores the method’s potential as a precise, radiation-free tool for evaluating spinal biomechanics in both clinical and research settings.

This study addresses key limitations of previous research by employing longer walking distances on solid ground and including participants with back pain, thereby enhancing the clinical relevance of the assessment. Future research should explore the broader application of this method in other clinically significant contexts, further expanding its utility and contribution to dynamic spinal assessments.

## Figures and Tables

**Figure 1 sensors-25-01647-f001:**
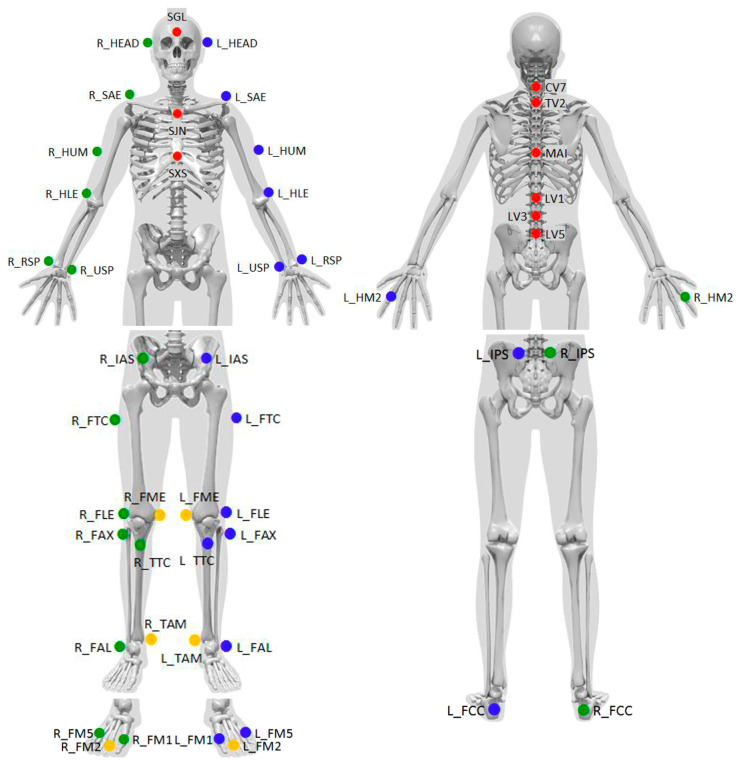
Placement positions of 49 reflective markers based on the “Qualisys PAF package: Instituti Ortopedici Rizzoli (IOR)”. This standard image from Qualisys illustrates marker placement as defined in the official protocol.

**Figure 2 sensors-25-01647-f002:**
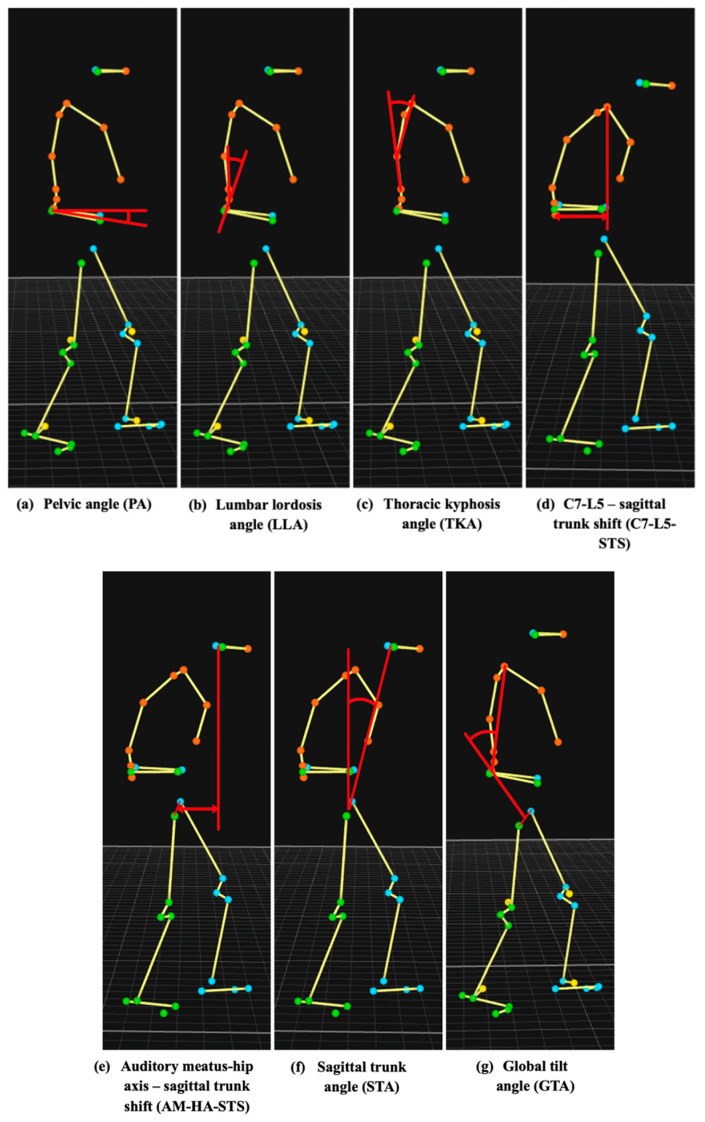
A 2D schematic representation of the measured parameters.

**Figure 3 sensors-25-01647-f003:**
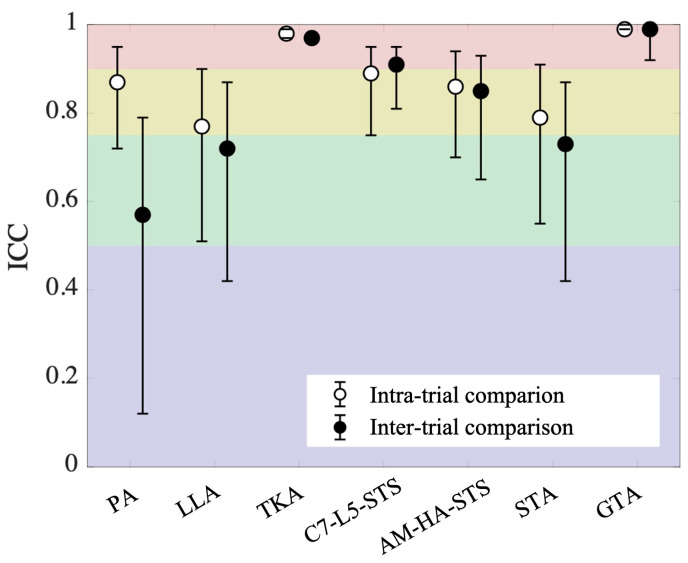
Intraclass correlation coefficients (ICC) with 95% confidence intervals for the measured parameters obtained during two consecutive right or left steps (intra-trial comparison) and across two different trials (inter-trial comparison). The red, yellow, green, and purple shaded areas indicate excellent, good, moderate, and poor test–retest reliability, respectively.

**Figure 4 sensors-25-01647-f004:**
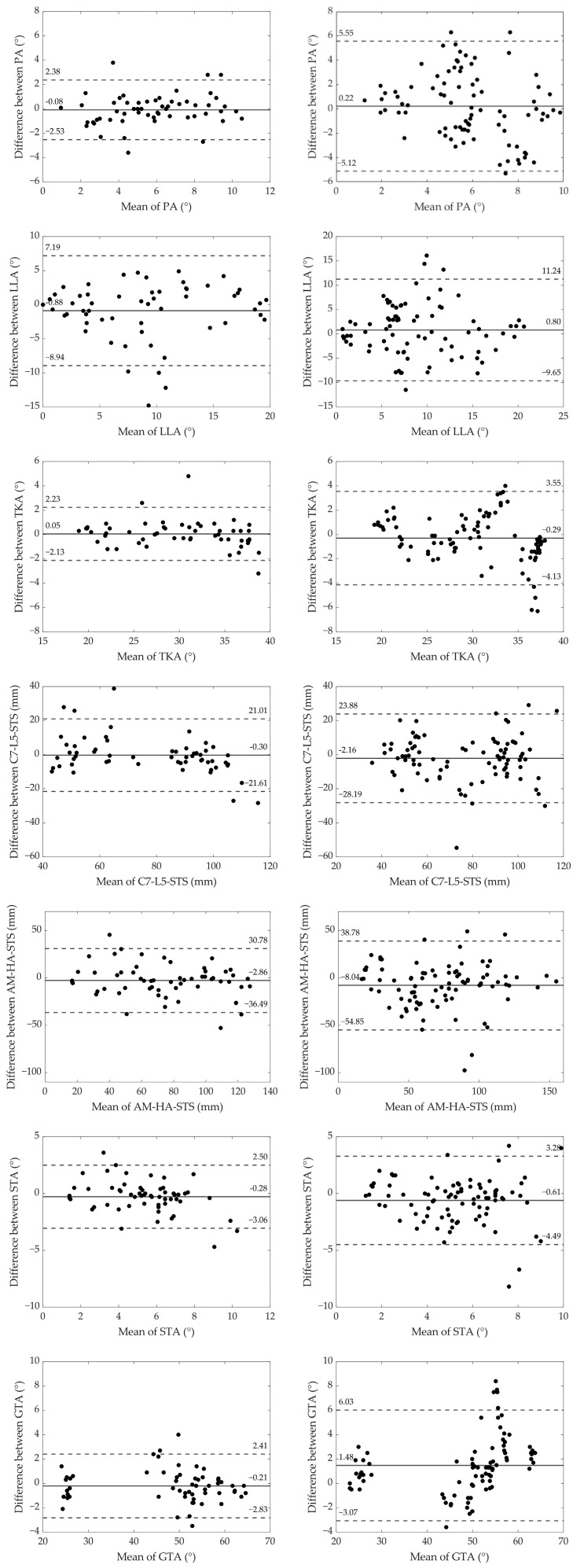
Bland–Altman plots comparing test and retest values for the measured parameters. Intra-trial comparisons (between two consecutive right or left steps) are displayed on the **left**, while inter-trial comparisons (across two different trials) are shown on the **right**. The solid line represents the mean difference between the two measurements, and the dashed lines indicate the limits of agreement (mean difference ± 1.96 times the standard deviation).

**Table 1 sensors-25-01647-t001:** Demographic and clinical characteristics of enrolled participants.

	*n* = 10
Age (years)	65.6 ± 5.6 (range: 56–73)
Sex (male/female)	1:9
Height (cm)	163.1 ± 8.7
Body mass (kg)	73.5 ± 15.4
Body mass index (BMI) (kg/m^2^)	27.5 ± 4.4
VAS during walking	4.0 ± 2.1
Sagittal vertical axis (SVA) (mm)	20.4 ± 27.3
Pelvic incidence-lumbar lordosis mismatch (PI-LL) (°)	0.5 ± 9.9
Participants with sagittal imbalance (SVA > 40 mm or PI-LL > 10°)	4

**Table 2 sensors-25-01647-t002:** Mean values and standard deviations of the measured parameters for all participants across both trials: (**a**) pre-walk static measurements; (**b**) post-walk static measurements; (**c**) initial-walk dynamic measurements for each step; (**d**) mid-walk dynamic measurements for each step; (**e**) end-walk dynamic measurements for each step.

(**a**)								
**Pre-Walk**	**Trial 1**				**Trial 2**			
PA (°)	4.4 ± 2.9				4.5 ± 3.3			
LLA (°)	8.9 ± 8.3				8.3 ± 6.5			
TKA (°)	29.3 ± 6.0				28.9 ± 6.5			
C7-L5-STS (mm)	24.3 ± 27.7				24.9 ± 27.4			
AM-HA-STS (mm)	70.5 ± 34.3				71.7 ± 32.7			
STA (°)	5.0 ± 2.1				5.0 ± 1.6			
GTA (°)	41.3 ± 13.0				40.4 ± 12.7			
(**b**)								
**Post-Walk**	**Trial 1**				**Trial 2**			
PA (°)	4.0 ± 2.9				3.5 ± 3.4			
LLA (°)	9.0 ± 7.6				9.0 ± 6.1			
TKA (°)	29.1 ± 6.4				29.0 ± 7.1			
C7-L5-STS (mm)	26.9 ± 24.4				26.0 ± 24.5			
AM-HA-STS (mm)	72.3 ± 31.7				63.3 ± 31.2			
STA (°)	5.1 ± 1.7				4.4 ± 1.8			
GTA (°)	39.8 ± 12.8				39.6 ± 11.9			
(**c**)								
**Initial-Walk**	**Trial 1**				**Trial 2**			
	Step 1	Step 2	Step 3	Average	Step 1	Step 2	Step 3	Average
PA (°)	5.9 ± 1.8	6.5 ± 2.1	6.0 ± 2.4	6.1 ± 2.0	5.9 ± 3.4	5.7 ± 2.9	6.6 ± 2.3	6.1 ± 2.8
LLA (°)	10.9 ± 6.5	9.3 ± 6.3	10.6 ± 6.1	10.3 ± 6.1	8.2 ± 6.2	7.7 ± 6.0	9.4 ± 6.5	8.4 ± 6.0
TKA (°)	30.2 ± 5.8	30.2 ± 6.0	30.2 ± 6.3	30.2 ± 5.8	30.5 ± 6.1	30.5 ± 6.2	30.6 ± 6.9	30.5 ± 6.2
C7-L5-STS (mm)	79.9 ± 21.6	76.3 ± 27.0	74.2 ± 32.9	76.9 ± 26.4	79.1 ± 19.0	75.6 ± 23.6	77.1 ± 29.3	77.3 ± 23.3
AM-HA-STS (mm)	73.7 ± 35.1	69.0 ± 41.3	71.1 ± 43.4	71.2 ± 38.6	79.6 ± 30.9	75.5 ± 39.3	77.6 ± 42.6	77.5 ± 36.4
STA (°)	5.4 ± 2.3	5.0 ± 2.7	5.2 ± 3.1	5.2 ± 2.6	5.8 ± 2.0	5.5 ± 2.7	5.7 ± 3.1	5.6 ± 2.5
GTA (°)	49.1 ± 13.0	48.4 ± 13.4	48.7 ± 14.1	48.7 ± 13.0	47.4 ± 12.4	47.1 ± 12.3	47.2 ± 12.9	47.2 ± 12.1
(**d**)								
**Mid-Walk**	**Trial 1**				**Trial 2**			
	Step 1	Step 2	Step 3	Average	Step 1	Step 2	Step 3	Average
PA (°)	5.6 ± 2.4	5.9 ± 2.2	5.7 ± 2.5	5.8 ± 2.3	5.4 ± 2.8	5.6 ± 3.0	5.5 ± 3.0	5.5 ± 2.8
LLA (°)	8.1 ± 6.0	7.2 ± 4.4	8.8 ± 4.9	8.0 ± 5.0	7.2 ± 6.1	8.5 ± 5.7	8.7 ± 6.6	8.1 ± 5.9
TKA (°)	29.7 ± 5.8	29.9 ± 6.0	30.0 ± 6.4	29.8 ± 5.9	29.7 ± 6.2	29.8 ± 6.4	29.9 ± 6.7	29.8 ± 6.2
C7-L5-STS (mm)	76.0 ± 22.8	73.6 ± 19.8	76.5 ± 23.5	75.3 ± 21.3	75.0 ± 23.6	76.9 ± 22.8	77.9 ± 22.8	76.5 ± 22.3
AM-HA-STS (mm)	69.2 ± 30.7	63.5 ± 32.6	73.1 ± 25.7	68.5 ± 29.2	73.4 ± 29.0	72.3 ± 35.6	77.1 ± 27.7	74.2 ± 30.0
STA (°)	5.0 ± 1.8	4.5 ± 1.6	5.4 ± 1.5	5.0 ± 1.6	5.3 ± 1.7	5.2 ± 2.0	5.6 ± 1.7	5.3 ± 1.7
GTA (°)	48.4 ± 13.4	48.7 ± 13.2	48.7 ± 14.4	48.6 ± 13.2	46.7 ± 12.8	47.1 ± 12.6	46.7 ± 13.1	46.8 ± 12.4
(**e**)								
**End-Walk**	**Trial 1**				**Trial 2**			
	Step 1	Step 2	Step 3	Average	Step 1	Step 2	Step 3	Average
PA (°)	5.5 ± 2.5	6.1 ± 1.8	5.4 ± 1.7	5.7 ± 2.0	5.9 ± 3.0	5.4 ± 3.0	6.0 ± 2.8	5.8 ± 2.8
LLA (°)	9.7 ± 5.1	9.2 ± 6.0	11.3 ± 7.2	10.0 ± 5.9	9.3 ± 6.8	8.8 ± 6.1	8.6 ± 6.1	8.9 ± 6.1
TKA (°)	29.8 ± 5.8	29.8 ± 5.8	29.5 ± 5.9	29.7 ± 5.6	30.5 ± 6.4	30.5 ± 6.9	29.9 ± 7.5	30.3 ± 6.7
C7-L5-STS (mm)	74.7 ± 20.9	75.8 ± 21.7	74.4 ± 21.8	75.0 ± 20.7	80.1 ± 20.2	80.8 ± 25.1	79.5 ± 29.9	80.2 ± 24.2
AM-HA-STS (mm)	67.6 ± 33.7	64.3 ± 40.2	63.0 ± 31.9	65.0 ± 34.3	76.8 ± 32.5	79.2 ± 42.8	78.1 ± 40.9	78.0 ± 37.5
STA (°)	4.9 ± 2.0	4.6 ± 2.2	4.5 ± 1.7	4.6 ± 1.9	5.6 ± 2.1	5.8 ± 2.9	5.8 ± 3.1	5.7 ± 2.6
GTA (°)	48.1 ± 13.2	48.4 ± 13.4	48.6 ± 14.1	48.3 ± 13.1	47.0 ± 12.3	47.4 ± 12.7	47.0 ± 13.7	47.2 ± 12.4

**Table 3 sensors-25-01647-t003:** Test–retest reliability of the walking trials presented as ICC values. The table compares the measured parameters between two consecutive right or left steps (intra-trial comparison, left) and across two separate trials (inter-trial comparison, right).

	Intra-Trial Comparison	Inter-Trial Comparison
PA	0.87 (0.72–0.95)	0.57 (0.12–0.79)
LLA	0.77 (0.51–0.90)	0.72 (0.42–0.87)
TKA	0.98 (0.97–0.99)	0.97 (0.96–0.98)
C7-L5-STS	0.89 (0.75–0.95)	0.91 (0.81–0.95)
AM-HA-STS	0.86 (0.70–0.94)	0.85 (0.65–0.93)
STA	0.79 (0.55–0.91)	0.73 (0.42–0.87)
GTA	0.99 (0.99–1.00)	0.99 (0.92–1.00)

**Table 4 sensors-25-01647-t004:** Differences in distances between specific marker pairs across both trials, measured in centimeters. Color coding indicates the range of differences: green (0–0.9 cm), purple (1.0–1.9 cm), blue (2.0–2.9 cm), and red (≥3.0 cm).

Markers Placement/Participant	1	2	3	4	5	6	7	8	9	10	Average
Head of the fibula-lateral malleolus—Left	0.1	1.2	1.1	2.3	0.1	4.3	0.7	0.2	0.9	0.6	1.2
Head of the fibula-lateral malleolus—Right	0.7	1.4	0.6	1.1	0.5	3.8	0.1	0.4	0.6	0.2	0.9
Greater trochanter-femoral lateral epicondyle—Left	1.3	1.0	0.8	0.5	1.4	4.4	1.8	0.6	1.5	0.3	1.4
Greater trochanter-femoral lateral epicondyle—Right	0.0	0.1	3.0	0.7	1.8	4.0	2.2	0.7	2.1	0.4	1.5
Anterior superior iliac spine left-right	0.3	0.7	3.1	0.3	2.6	0.3	0.6	0.7	0.3	1.1	1.0
Posterior superior iliac spine left-right	0.2	2.0	1.1	1.2	2.2	1.1	0.4	1.0	1.4	0.6	1.1
Acromial edge-humeral lateral epicondyle—Left	0.0	0.5	1.3	0.3	1.9	0.9	0.5	0.6	0.1	0.8	0.7
Acromial edge-humeral lateral epicondyle—Right	0.8	0.4	1.9	0.4	1.6	0.1	0.2	0.1	0.4	1.2	0.7
Humeral lateral epicondyle-radial styloid process—Left	1.2	0.6	0.2	0.7	0.7	0.5	0.5	0.1	1.3	1.4	0.7
Humeral lateral epicondyle-radial styloid process—Right	0.6	0.7	1.1	1.2	1.1	0.2	1.6	0.8	1.5	1.0	1.0

## Data Availability

The data supporting the findings of this study are available from the corresponding author upon reasonable request.

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
