# Peer review of "Reliability of Spino-Pelvic and Sagittal Balance Parameters Assessed During Walking in Patients with Back Pain"

_sensors, 2025, doi:10.3390/s25061647_

Round 1
Reviewer 1 Report
Comments and Suggestions for Authors
This study explores the reliability of spinal-pelvic and sagittal balance parameters during motion. The volunteer group targeted by the study consists of elderly patients with back pain characteristics. The reliability assessment covers the primary existing measurement parameters, with evaluation methods including comparisons of data from different trials of the same volunteer and comparisons of experimental data between different volunteers. Research on diseases based on motion capture systems has been widely conducted, with reliability being a major challenge faced by such studies. This research provides an important reference for selecting spinal-pelvic and sagittal balance parameters based on motion capture systems. However, the manuscript has the following issues that need further revision.
1. Page 4 Line 110
It is recommended to include a complete list of the names of the markers for reference by researchers from non-medical fields.
2. Page 6 Line 183
The manuscript provides detailed explanations of the parameter measurements; however, Figure 2 only provides a two-dimensional illustration of the measurements without specifying the exact measurement tools used for these parameters. Were all of the parameters automatically measured or calculated using the optoelectric Motion Capture System?
3. Page 7 Line 212
The data provided in the manuscript represent the mean values of measurements from each volunteer. Did the authors perform any individual analysis of each volunteer's data? Given that Table 1 shows substantial measurement differences between volunteers, would basing the analysis entirely on mean data potentially obscure some of the variability information?
4. Page 7 Line 219 “In contrast, PA displayed the greatest variability, with pre-walk values of 4.4° ± 2.9° and 4.5° ± 3.3°, and post-walk values of 4.0° ± 2.9° and 3.5° ± 3.4° for Trials 1 and 2, respectively.”
The data in Table B suggest that the variability between different trials seems to be higher for STA. The authors should further confirm this observation.
5. Page 12 Line 287
An explanation of the measurement method for the marker distances between different trials needs to be added.
6. Page 12 Line 289
It is recommended to include recent relevant literature to further support the conclusions of the study.
Reviewer 2 Report
Comments and Suggestions for Authors
This study evaluated the reliability of spinopelvic and sagittal balance parameters measured during walking in patients with back pain. Ten patients aged 56-73 years completed a six-minute walking assessment while being monitored with an optoelectric motion captures system. Forty-nine reflective markers were placed to measure the spinopelvic and sagittal balance parameters. The results showed excellent reliability for thoracic kyphosis angle, C7-L5 sagittal trunk shift, and global tilt angle; good reliability for auditory meatus-hip axis sagittal trunk shift; and moderate reliability for pelvic angle, lumbar lordosis angle, and sagittal trunk angle. While this study has important clinical implications, several factors remain to be considered:
General Comments
1. The authors state that “Sagittal balance is a complex phenomenon that requires comprehensive diagnostics with standardized standing full spine X-rays providing only static measurements.” In this paper they advocate for a “more detailed, dynamic evaluation of sagittal balance,” using a marker-based motion capture system during walking. However, the authors don’t incorporate any of the dynamic changes that are occurring during the gait cycle into their analysis.
2. The assessments of the spinal motion used in this study are singular-plane measurements. However, the spinal column has a complex structure that moves in three-dimensional space during physical activity. Therefore, measuring planar motions, whether with a static full-standing x-ray or with a motion capture system during gait, is not a true representation of spinal motion.
3. The study evaluated 10 subjects with back pain. However, the level or severity of the back pain is not provided. Further, if the back pain is a result of disc degeneration, the paper lacks quantification of the degree of degeneration using standard techniques.
4. How many subject records were evaluated and how many subjects were contacted to get the 10 subject who did participate in the study?
Specific Comments:
Page 1, line 23: Was the study aim to establish or assess reliability?
Page 2, line 65: Please provide references describing the existing approaches.
Reviewer 3 Report
Comments and Suggestions for Authors
The study is dedicated to walking biomechanics, nice written, clear for understanding, but still contain several issues:
1.The end of the introduction please add details, sounds very general
2.Methods, chartflow for inclusion\exclusion procedure is needed
3.For tab 1, and for the first 5 parameters are shown please provide representability of the sample, by comparing with literature.
4. Fig 1 to complete the real photo taken during the study, subject must be anonymized
5.Numerize fig2 (by subfigures), provide refs to these subfigs at 2.4 items
6.More detail explanation of ICC is needed in the text
7.Fig4 lines 4 and 7, what could authors say about two subgroups? In my opinion cluster-analysis is needed
8.How exactly the data obtained could be implemented for futher research and clinical app-s? add for abstract and discussion sections.
9.Key words extend : smth with automated video assimilation (on authors preference).
Round 2
Reviewer 2 Report
Comments and Suggestions for Authors
The authors have submitted a revised manuscript. Most of the reviewer’s comments have been addressed. However, two comments still need to be addressed:
- The paper title indicates that the reliability of the Spino Pelvic and Sagittal Balance Parameters will be assessed during walking. As written, this study does not report on any dynamic changes throughout the gait cycle. Without this analysis, the study is no different than a static assessment. The authors need to report on the variability of the Spino Pelvic and Sagittal Balance Parameters throughout the gait cycle.
- Since this is study performed on patients with back pain, the author need to report the back pain severity for the patient in the study in order to provide context for the study.
